# PriMAT: Robust multi-animal tracking of primates in the wild

Richard Vogg[1,2], Matthias Nuske[3], Marissa A. Weis[1], Timo Lüddecke[1], Elif Karakoç[2], Zurna Ahmed[4], Sofia M. Pereira[5,6], Suchinda Malaivijitnond[7,8], Suthirote Meesawat[7], Derek Murphy[9,10], Julia Fischer[9,10,11,12], Florentin Wörgötter[3,11,12], Peter M. Kappeler[2,11,13], Alexander Gail[4,11,12,14], Julia Ostner[5,6,11], Oliver Schülke[5,6,11], Claudia Fichtel[2,11], Alexander S. Ecker[1,11,12,15]*

1 Institute of Computer Science and Campus Institute Data Science, University of Göttingen, Göttingen Germany, 2 Behavioral Ecology and Sociobiology Unit, German Primate Center, Leibniz Institute for Primate Research, Göttingen, Germany, 3 Department for Computational Neuroscience, Third Physics Institute, University of Göttingen, Göttingen, Germany, 4 Sensorimotor Group, German Primate Center, Leibniz Institute for Primate Research, Göttingen, Germany, 5 Behavioral Ecology Department, University of Göttingen, Göttingen, Germany, 6 Social Evolution in Primates Group, German Primate Center, Leibniz Institute for Primate Research, Göttingen, Germany, 7 National Primate Research Center of Thailand, Chulalongkorn University, Saraburi, Thailand, 8 Department of Biology, Faculty of Science, Chulalongkorn University, Bangkok, Thailand, 9 Cognitive Ethology Laboratory, German Primate Center, Leibniz Institute for Primate Research, Göttingen, Germany, 10 Department for Primate Cognition, Johann-Friedrich-Blumenbach Institute, University of Göttingen, Göttingen, Germany, 11 Leibniz ScienceCampus, German Primate Center, Leibniz Institute for Primate Research, Göttingen, Germany, 12 Bernstein Center for Computational Neuroscience, University of Göttingen, Göttingen, Germany, 13 Department of Sociobiology/Anthropology, University of Göttingen, Göttingen, Germany, 14 Georg-Elias-Müller-Institute of Psychology, University of Göttingen, Göttingen, Germany, 15 Max Planck Institute for Dynamics and Self-Organization, Göttingen, Germany

* ecker@cs.uni-goettingen.de

## Abstract

Detection and tracking of animals is an important first step for automated behavioral studies using videos. Animal tracking is currently done mostly using deep learning frameworks based on keypoints, which show remarkable results in lab settings with fixed cameras, backgrounds, and lighting. However, multi-animal tracking in the wild presents several challenges such as high variability in background and lighting conditions, complex motion, and occlusion. We propose PriMAT, an approach for tracking nonhuman primates in the wild. PriMAT learns to detect and track primates and other objects of interest from labeled videos or single images using bounding boxes instead of keypoints. Using bounding boxes significantly facilitates data annotation and robustness. Our one-stage model is conceptually simple but highly flexible, and we add a classification branch that allows us to train individual identification. To evaluate the performance of our approach, we applied it in two case studies with Assamese macaques (*Macaca assamensis*) and redfronted lemurs (*Eulemur rufifrons*) in the wild. Additionally, we show transfer to other settings and species, particularly, Barbary macaques (*Macaca sylvanus*), Guinea baboons (*Papio papio*), chimpanzees (*Pan troglodytes*), and gorillas (*Gorilla spp.*). We show that with only

**Data availability statement:** All images for model training, all videos for validation and testing, as well as qualitative model output videos are available here: https://data.goettin-gen-research-online.de/dataverse/sfb1528_z02 (DOI: https://doi.org/10.25625/CMQY0Q) All codes are available on Github: https://github.com/ecker-lab/PriMAT-tracking.

**Funding:** This project was funded by the Deutsche Forschungsgemeinschaft (DFG, German Research Foundation) via project number 454648639 – SFB 1528, project number 254142454 – GRK 2070 and project number 502807174 – RTG 2906. We acknowledge funding by the Leibniz Association through an Audacity Grant from the Leibniz ScienceCampus Primate Cognition (W45/2019 – Strategische Vernetzung). The funders had no role in study design, data collection and analysis, decision to publish, or preparation of the manuscript.

a few hundred frames labeled with bounding boxes, we can achieve robust tracking results. Combining these results with the classification branch for the lemur videos, the lemur identification model shows an accuracy of 84% in predicting identities. Our approach presents a promising solution for accurately tracking and identifying animals in the wild, offering researchers a tool to study animal behavior in their natural habitats. Our code, models, training images, and evaluation video sequences are publicly available at https://github.com/ecker-lab/PriMAT-tracking, facilitating their use for animal behavior analyses and future research in this field.

## Introduction

Automated analysis of actions and social interactions of free-ranging animals is essential for advancing behavioral research, enabling the study of complex behaviors in natural settings. Video recordings are a valuable tool for studying animal behavior, but traditional manual annotation of videos is time-consuming and labor-intensive. In the context of primate behavior research, large-scale annotated datasets have recently enabled the development and training of deep learning based models [1–8]. There are models which show promising results on several computer vision tasks in the wild, such as individual identification [9–12], pose estimation [13,14], object detection [15], tracking [16,17] and action classification [18]. However, the vast majority of frameworks used to analyze animal behavior are designed to work in laboratory settings and rely on pose estimation as an intermediate representation [19,20]. Some of the most used frameworks for such markerless pose estimation are DeepPoseKit [21], DeepLabCut [22,23], SLEAP [24], and TRex [25]. They allow for multi-animal pose tracking and have user-friendly interfaces that help with annotation and inference. The introduction of these frameworks has contributed to great progress in the area of behavioral analysis [26–28].

As valuable as these tools are in the lab, their utility for applications of tracking animals in the wild is limited [19,29]. Recording animals in their natural habitat presents challenges such as occlusion, diverse backgrounds, and varying lighting conditions. Similar appearances between individuals and sudden, rapid movements add complexity to the task [30]. We propose a different approach as a step towards analyzing individual actions and social interactions in videos of wild primates. Instead of keypoints as an intermediate representation of the objects, we propose to detect bounding boxes for all animals and relevant objects in the scene (Fig 1).

We see three main reasons why object detection via bounding boxes should be a starting point to consider for many applications. First, currently, there are few successful applications of the above-mentioned keypoint-based frameworks on videos from the wild. Perez and Toler-Franklin [19] review CNN-based action recognition and pose-estimation pipelines for animal behavior and highlight that keypoint-based approaches are now standard for controlled lab recordings. They also explicitly list generalization to in-the-wild conditions and robustness to cluttered backgrounds as major open challenges. In the years after this survey, several methods were

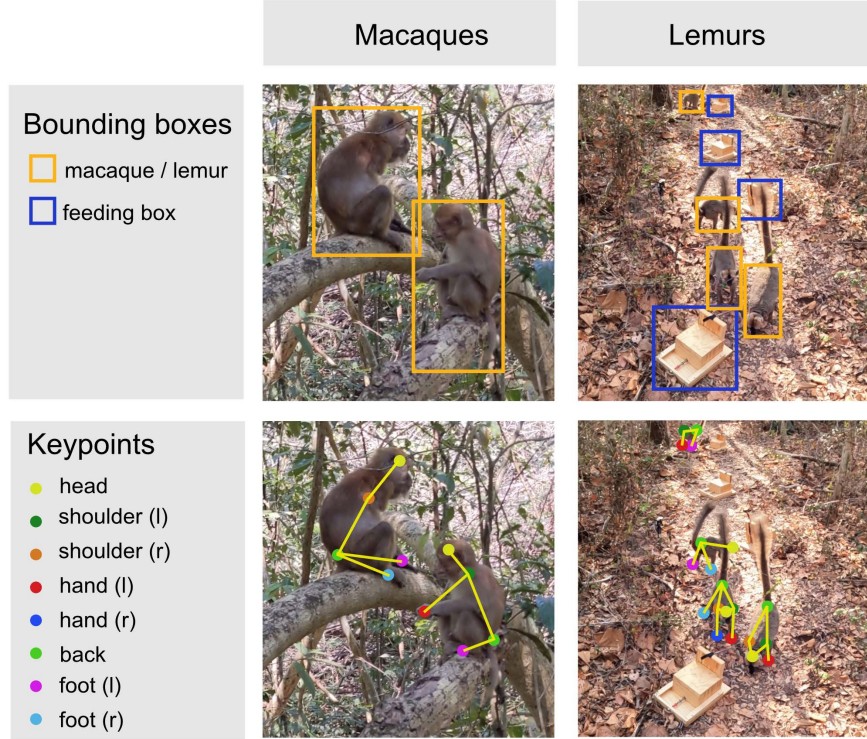

**Fig 1. Bounding box based detection vs. keypoint detection on different datasets.** Bounding box based tracking has advantages over keypoint tracking when used as a starting point for automated behavior analysis, such as reduced labeling time, higher tracking robustness for videos from the wild, and universal extensibility to other objects of interest.

developed for keypoint tracking in the wild, requiring high annotation efforts and low generalization to unseen videos. DeepWild showed that DeepLabCut can learn from around 2000 labeled frames from videos of animals in the wild; however, the results on unseen videos leave room for improvement [13]. Similarly, Duporge et al. [31] train a well-functioning tracking model for keypoints of zebras by annotating 1920 frames and getting good results only on the videos from which they annotated. A study conducted on horse images showed the limited capability of keypoint-based models to generalize to new, unseen horses [29]. For humans, pose tracking with variable backgrounds works [32,33]. The difference here is that diverse, large-scale datasets exist and substantial human labor has been put into annotating them [34,35]. For most animal species, neither large-scale datasets with broad enough coverage exist, nor has an equivalent amount of labor been invested into annotations. Second, using an object-detection framework built on bounding boxes may be preferable because keypoints require significantly more time to annotate than bounding boxes. Wiltshire et al. [13] reported that they needed around two hours to label 18 keypoints for all animals in 25 frames (less than 175 individuals, as videos were chosen to have not more than seven individuals present). We annotated around 200 frames (roughly 1500 bounding boxes) in the same time, which is roughly an order of magnitude more individuals. Third, many questions in animal behavior research, especially in the wild, simply may not require keypoints but instead focus on actions and interactions which can directly be extracted from the image information inside the bounding box.

In this paper, we propose PriMAT, a conceptually simple **Pri**mate **M**ulti-**A**nimal **T**racking approach. It builds on the Fair-MOT model [36], which is trained on individual images and thus does not require entire video sequences to be labeled. This greatly reduces annotation effort, as only selected frames need to be annotated instead of complete videos, while the trained model can be applied to videos during inference. Due to the existing datasets [34,37] and benchmarks [38–40] for

multi-object tracking, FairMOT, like most tracking models, was designed for tracking human pedestrians. However, there are large differences between humans and nonhuman primates, as well as among nonhuman primate species, in appearance, motion patterns, and acceleration. Pedestrians in city scenes are generally easily distinguishable from the background [40]. This makes them easier to detect and track than nonhuman primates in their natural habitat, especially when the primates are in non-standard poses or highly occluded. Furthermore, their locomotion differs fundamentally from the relatively consistent, bipedal gait of humans. These differences pose challenges for applying standard pedestrian tracking models to primate behavior studies in naturalistic settings.

Our contributions are fourfold: 1) We adapted the tracking strategy of PriMAT to suit the appearance and motion of monkeys and lemurs, and present use-cases with highly variable backgrounds and lighting conditions. 2) We release two annotated video benchmarks for multi-animal tracking, including videos and frame-level labels for redfronted lemurs and Assamese macaques, comprising 24 videos of 12 seconds duration for each species. 3) We show that our bounding box tracking approach outperforms a keypoint tracking model in the wild. 4) We added a branch that can learn classification tasks on top of the tracking results and used it to identify individual lemurs.

## Materials and methods

### Multi-animal tracking for primates

Transfer learning is an effective way to reduce annotation effort by starting with a base model pretrained on a large dataset and fine-tuning it with a few hundred annotated frames from the domain of application. We experimented with four pretraining datasets (see below) and annotated 500 frames from videos of lemurs and macaques to demonstrate the applicability of the base model to these species (Fig 2A).

The PriMAT architecture is structured into three parts (Fig 2B): *(1)* a convolutional backbone used for feature extraction, *(2)* multiple heads that perform different tasks for tracking, and *(3)* a classification branch. More detailed information about backbone, heads and association methodology can be found in S1 Appendix.

We adapted the tracking strategy in the following manner to account for the differences in appearance and motion between pedestrians and nonhuman primates. First, we considerably lowered the threshold for potential detections, compared to the strategy in FairMOT. This adjustment was necessary to maintain track continuity, especially in cases where the animals were moving. As low detection thresholds introduce potential false positive detections, we added a threshold to prevent new, unmatched boxes from starting a new track if they have too much overlap with already existing tracks. Additionally, we removed detections less than five pixels away from the image border, as most false positives were found in these areas. The original FairMOT model uses only learned appearance features for association, relying on intersection-over-union (IoU) for unmatched tracks and detections. Nonhuman primates have much less inter-individual variation in appearance; thus, for our use cases, location is a more important predictor than appearance. For this reason, we use a linear combination of appearance and location similarity in one step to calculate association.

Tracking models typically use a Kalman filter to predict the location of objects from frame to frame. The Kalman filter assumes linear motion of objects and predicts the position of an object based on the velocity over the previous frames, which may not be appropriate for tracking nonhuman primates. For instance, when jumping, a lemur moves with high velocity, which then drops close to zero once the animal lands. This kind of movement can cause problems when calculating the association (Fig 3). To avoid problems with fast motion changes, we added a conditional second stage to the association procedure. If there is no match at the location predicted by the Kalman filter, we additionally check for matches at the location of the last detection.

### Pretraining

The goal of the pretraining is to use already existing annotated datasets that are either general-purpose – covering a wide range of visual categories – or close to the domain of application. We compared ImageNet [43],

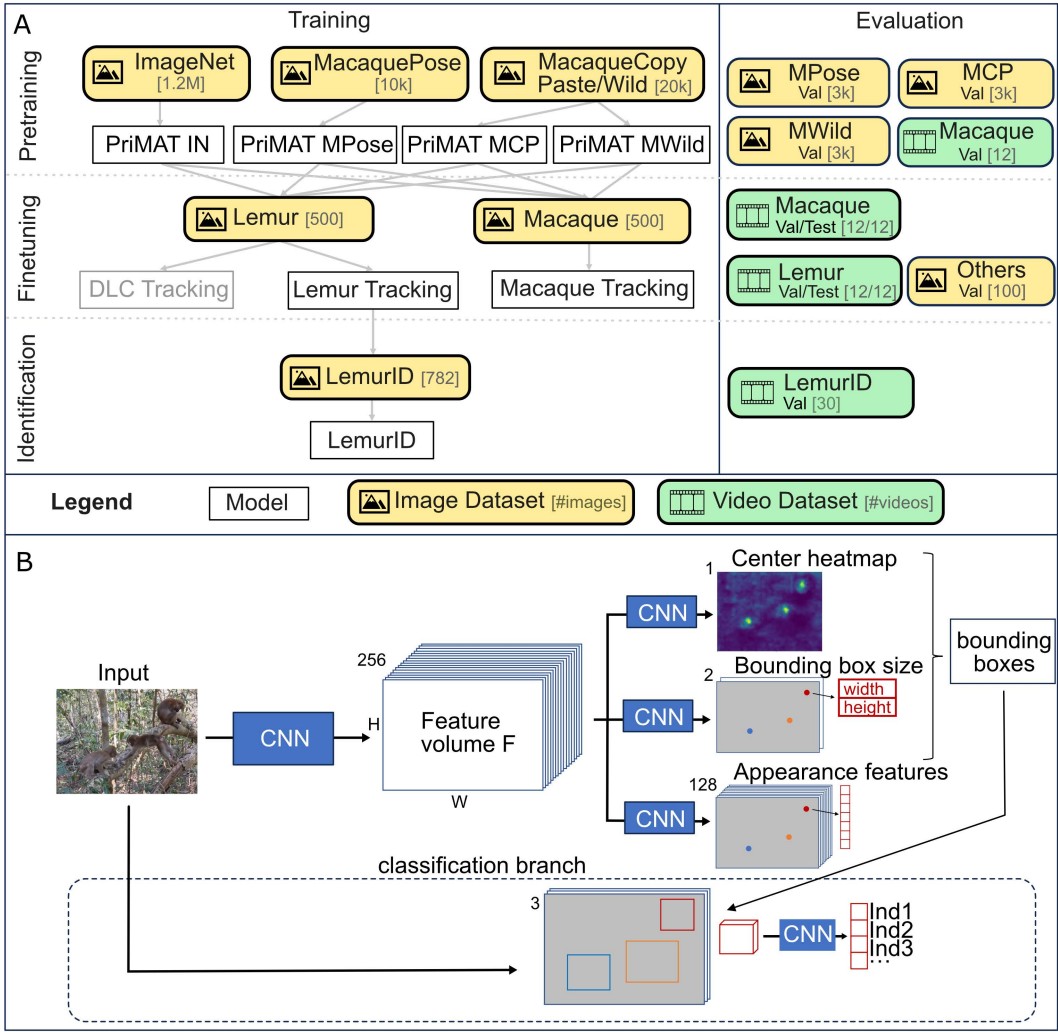

**Fig 2. (A) Structure of our datasets and experiments.** All models were trained with image datasets, while the performance was evaluated primarily on videos. We compared different datasets for pretraining, and applied them to two use cases: videos of redfronted lemurs and Assamese macaques. We compared our bounding box based approach with DeepLabCut [23]. Finally, we demonstrated the use of our additional classification branch for individual identification. **(B)** Overview of the model architecture: Input images are processed by a Convolutional Neural Network (CNN, in our case HRNet32 [41]) into a feature volume. Different heads learn tracking-related tasks using two-layer CNNs. We omitted the offset head for simplicity. Afterwards, individuals are cut out and processed by the classification branch for individual identification. We used ResNet18 [42] as the CNN for individual recognition.

MacaquePose [3] and two augmented versions of MacaquePose for pretraining. ImageNet is a standard dataset for pretraining, containing over 1.2 million images from a variety of object classes. MacaquePose consists of 13,083 images of macaques, mostly taken from zoos, and is completely annotated with instance segmentation masks and keypoints. We constructed bounding boxes from the extreme points of each segmentation mask. Although MacaquePose contains images of animals in outdoor situations, the dataset differs from our forest video domain. To enhance generalization and increase background diversity, we applied copy-paste augmentation to MacaquePose using the labeled instance masks (Fig 6A). We created two augmented datasets: MacaqueCopyPaste, in which individuals were pasted onto ImageNet backgrounds (Fig 6A), and MacaqueCopyPasteWild, where backgrounds were taken from the Phu Khieo Wildlife Sanctuary in Thailand. Previous work has shown that copy-paste augmentation

Predicted bounding boxes

Kalman filter estimates for bounding box centers

**Fig 3. Problem during a jump: The Kalman filter predicts the location of the lemur track based on past velocity.** The detection in the last frame cannot be matched to the lemur track. Top row: Detected bounding boxes and assigned tracks identified by the color of the bounding box. The track of the jumping lemur is lost in the last frame (black) and a new track is instantiated (green). Bottom row: Kalman filter predictions for the expected position of the lemur. The Kalman filter assumes linear motion and predicts a different position when the jump ends (black). The newly instantiated track starts with default parameters for the Kalman filter prediction (green).

improves model performance in instance segmentation [44,45] and object detection [46,47], offering an efficient way to generate realistic training images across diverse backgrounds. We split the three datasets MacaquePose, MacaqueCopyPaste, and MacaqueCopyPasteWild into 80% training data and 20% test data and trained one model on each of them. We evaluated each model on the test set of all three datasets using mean Average Precision (mAP), a classical metric for object detection tasks [37,48].

## Assamese macaques

Data on Assamese macaques were collected as part of a long-term project on the behavioral ecology of wild Assamese macaques in the Phu Khieo Wildlife Sanctuary, North East Thailand [49]. The study groups were all habituated to the presence of human observers for a long-term investigation that started in 2005, and animals did not show any signs of disturbance during focal animal follows or video recording. As a general rule, human observers stay at 3-5m away from the animals at all times. At no time were the macaques fed, trapped, or handled in any other way. Data were collected on approximately 250 individually recognizable subjects of different age-sex classes organized in up to five large multi-male/multi-female social groups ranging in size from 40 to 73 individuals [50]. The habitat consists of dense hill evergreen forest, and the monkeys spend the vast majority of their time in the trees, reducing visibility and creating difficult video-recording conditions. The videos were hand-recorded with resolutions of 1920×1080 using a GoPro Hero 10 camera and 3840×2160 using a Google Pixel 4 camera. More than 300 video snippets have been obtained so far. The videos consist

of sequences that are challenging for a tracking model, as they can contain motion blur through camera or animal movement, severe occlusion, and high variability in backgrounds and lighting conditions.

We manually annotated the bounding boxes on 500 frames randomly sampled from the macaque videos. To extract meaningful samples, we used the model pretrained on MacaquePose, which yielded an approximate count of individuals in each video. Then, we sampled images from the macaque videos according to the macaque count, i.e., we chose more examples from crowded scenes and fewer from scenes with lone individuals. We used the software VoTT [51], a user-friendly image annotation tool to label training frames, and used tight modal bounding boxes, that is, boxes around the outermost visible pixels of each individual for all applications.

### Redfronted lemurs

We recorded videos during a social learning experiment in Kirindy Forest, Madagascar [52]. Cameras were placed before the lemurs came to the experiment (for the exact setup, please see the information below). Redfronted lemurs participated voluntarily in the experiments and were free to approach the cameras, which they occasionally did, or avoid the cameras. Hence, redfronted lemurs were not handled during image collection.

The experiment involved four groups of redfronted lemurs, with group sizes ranging from five to eight individuals. In each lemur group, we presented four feeding boxes in each experiment and repeated the experiments ten times over the course of three months. The duration of each experiment varied from 10 to 30 minutes. We placed eight GoPro Hero 10 cameras to record the interactions of individuals with the feeding boxes from different angles (Fig 4A). All cameras recorded videos with a resolution of and 30 frames per second. Four cameras were placed to record the opening mechanism of each box from a distance of one meter (Fig 4B). Two cameras mounted on nearby trees were used to capture a top-down view of pairs of boxes (Fig 4C). Additionally, we placed two cameras on tripods to capture an overview of the experimental set-up from two opposite sides (Fig 4D). Each video was treated independently by the model, without taking into account input from the other perspectives. We did, however, train one model to work well on all camera perspectives, instead of training three specialized models.

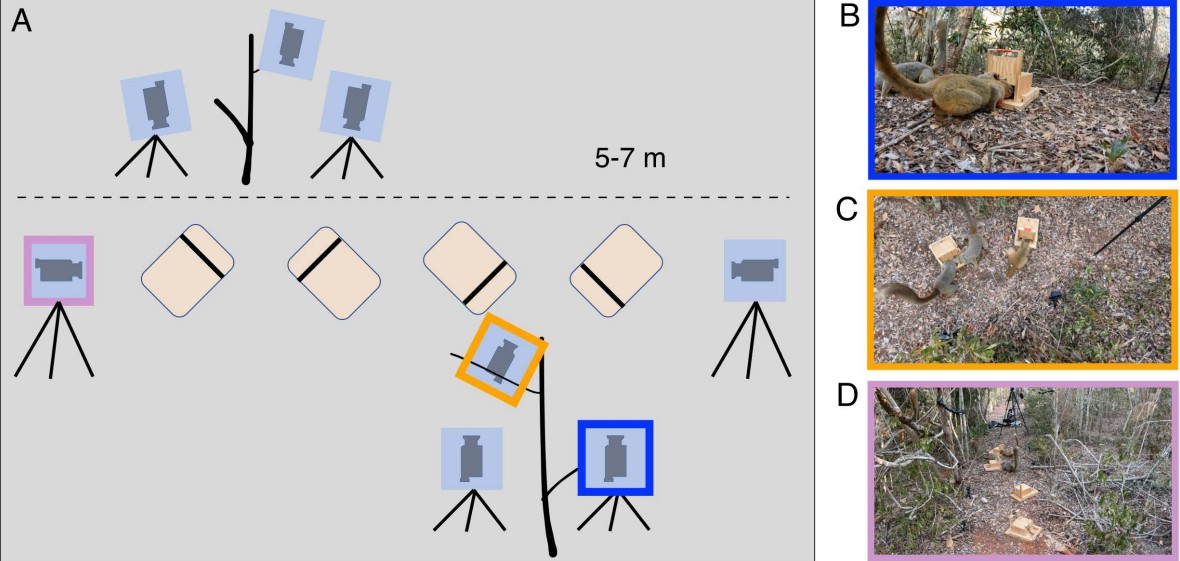

**Fig 4. (A) Setup for social learning experiments: Four feeding boxes were placed on the ground and eight cameras were filming from three different perspectives ((B) close, (C) top, (D) far).**

We recorded eight videos from different perspectives for each of the 40 experimental sessions. Additionally, we conducted pre-experiment sessions with open boxes so the lemurs could get used to the experimental setup. We used a subset of 50 videos from the pre-experiment sessions to obtain our training set. As the close-up view showed the most variability in lemur appearance, we used 24 videos from this view and 13 videos each from the far view and top views. In each video, we determined the longest segment that continuously contained at least one lemur, and randomly extracted 10 frames from each of these segments. The 500 resulting frames formed the training set. In these images, we followed the same procedure as for the macaques and annotated all lemurs by manually drawing the bounding boxes using VoTT. We decided to exclude the lemur tails from the bounding boxes, as they would otherwise occupy a relatively large proportion of the enclosed space and lead to fast changes in bounding box size and location from frame to frame. Additionally, we labeled all feeding boxes in the videos to showcase the multi-class capabilities of our model.

### Tracking evaluation

We followed the same evaluation strategy for both species. For each species, we annotated 12 sequences, each one 12 seconds long, frame-by-frame with 30 frames per second, to evaluate the tracking performance with different hyperparameters and settings. For the lemur experiment, we additionally annotated the feeding boxes with bounding boxes. Similarly, we annotated 12 sequences for each species of the same length to form a test set, which can be used for benchmarking against our method. Validation and test videos were taken from experiments that were not used to extract training images. For the macaques, the sequences had varying levels of environmental complexity, and each included some motion or occlusion. For the lemurs, we chose the same number of videos from each camera perspective. For quality control of the annotations used for evaluation, we assessed the consistency between the labels produced by two human annotators on a subset of three videos of the macaque dataset. We matched the annotated bounding boxes using Hungarian matching and calculated the average intersection over union across all boxes. The average IoU score was over 0.8 for each of the three videos.

We used the video annotation tool CVAT [53] to label the evaluation and test videos. CVAT offers the advantage of automatically interpolating bounding boxes between frames, reducing the need for manual annotation when consecutive frames remain unchanged. The tool has been successfully used in the past for animal annotation [30]. In our use cases, every primate in every frame was annotated if the annotator was able to recognize it as a primate in the still image. If a primate was temporarily occluded, it was labeled with the same ID after reappearing.

We evaluated the performance of all models with several metrics commonly used for multi-object tracking benchmarks. The most important metric is considered to be Higher Order Tracking Accuracy (HOTA) [54], as it gives equal importance to detection and association. Additionally, we report Multi-Object Tracking Accuracy (MOTA) [55] and IDF1 [56], two traditional metrics for easier comparison to other tracking models.

We compared the performance with the state-of-the-art video segmentation model SAM3 [57]. As the original model can only track one object class at a time, we run it twice for the lemur experiments, once for the lemurs and once for the feeding boxes. We convert the output into bounding box tracks and evaluate identically to our model.

### Transfer to other species

We trained a model jointly with the 500 images from lemur and 500 images from Assamese macaque videos to test its applicability to other settings or other primate species. We tested the model on videos of Barbary macaques (*Macaca sylvanus*) that were recorded as part of a decision-making experiment in Rocamadour, France [58], videos of Guinea baboons (*Papio papio*) ranging near the CRP Simenti Field Station in the Niokolo Koba National Park in Senegal [59], and videos from the PanAf500 dataset, which includes chimpanzees (*Pan troglodytes*) and gorillas (*Gorilla spp.*) [60]. For each dataset, we manually selected 100 frames and annotated the individuals using bounding boxes. We evaluated the detection performance with a 5-fold cross-validation. As we were operating with annotated images, and not videos, we

reported mean Average Precision (mAP) as a standard detection metric. We additionally reported AP50, a relaxed variant of mAP, where a bounding box counts as correctly predicted if it has an IoU of at least 0.5 with a ground truth bounding box. Models that score high on AP50 but low on mAP are good at finding the presence of an animal, but do not have perfectly precise bounding boxes.

### Bounding box and keypoint based detection

In light of the studies mentioned in the introduction suggesting that models for keypoint estimation in the wild struggle with generalization, we performed a small experiment. The goal of this experiment is a fair comparison between methods, helping practitioners determine which approach is more suitable for their setting. For most primate settings, directly applicable models are unavailable, requiring models to learn to generalize to a new domain. This requires data annotation. To ensure fairness, we allocated a fixed annotation time budget and assessed whether reliable performance on unseen videos could be achieved with the resulting annotated data.

We compared our lemur model with keypoint-based tracking by training a DeepLabCut model [23] to perform keypoint tracking on our lemur videos (Fig 8A). We allocated five hours for annotation – the same time required to label 500 frames with bounding boxes – and annotated 33 keypoints per individual in 100 frames. We had annotated 500 frames for the bounding box model by selecting ten frames from each of 50 videos. For the subset, we randomly selected two out of the ten annotated frames from each of the 50 videos, ensuring a comparable background variability.

We evaluated the accuracy of keypoint tracking on seven selected keypoints (nose, head, neck, spine, start tail, mid tail, end tail) by annotating additional validation frames from the same 50 videos used for extracting training images. For this, we followed the metric suggested in DeepLabCut, and calculated the root-mean squared error for each detected keypoint to the closest ground truth detection [23]. We used a simple metric to compare detection performance from the bounding box and the keypoint tracking approaches, as we only had bounding box ground truth labeled for complete video sequences. For each ground truth bounding box, we determined whether there was a predicted bounding box with an intersection-over-union (IoU) value of at least 0.5, a standard threshold to decide whether a predicted bounding box should be considered to represent a ground-truth bounding box based on the amount of overlap between them [48]. We then counted the number of predicted bounding boxes that overlapped sufficiently with each ground truth box (i.e., the number of correct detections). Similarly, we counted how often there were at least two ground-truth keypoints within a predicted bounding box. As the keypoints could be close to the box border, we also counted the keypoints that were within the 1.5× buffered bounding box. The value of 1.5 was chosen as it is a generous extension of the bounding box to allow even imprecisely detected keypoints to count towards a correct detection. For both approaches, we reported the proportion of correct detections out of the total number of ground truth bounding boxes as the recall value for each video.

Additionally, we calculated the precision of each model as the proportion of correct detections based on predicted keypoints or bounding boxes, respectively, out of the total number of predictions from each video. We reported the F1 score in both cases,

$$F1 = \frac{2 \times precision \times recall}{precision + recall}.$$

### Individual identification

To demonstrate how the tracking approach can be extended, we added a new branch and trained it to identify individual lemurs. This task is also often referred to as closed-set re-identification in the animal behavior literature. All lemurs wore collars with tags featuring a unique combination of different shapes and colors. These tags are also used by field researchers to distinguish individuals.

To train the individual identification model, we manually selected 350 frames from close-up video sequences of a single group comprising seven individuals. From 40 videos of this group, we extracted frames in which the collar of at least one individual was clearly visible. To ensure diversity in background and lighting conditions, a maximum of 25 frames were selected per video. As before, bounding boxes were annotated using the Visual Object Tagging Tool (VoTT), with each box labeled according to the individual's identity. For any lemurs present in the frames whose collars were not visible, the corresponding annotations were labeled as "Unsure". To evaluate identification performance on temporal tracks, we selected ten close-up camera videos and extracted three one-minute video sequences from each, resulting in a total of 30 sequences. Since the primary objective was to assess individual identification accuracy –rather than tracking robustness – we manually curated the sequences to exclude instances involving ID switches in the tracking output. An ID switch occurs when two distinct individuals are erroneously merged into a single track (e.g., due to occlusion or close proximity), which makes correct identification impossible, as there is no correct identification on track level. The final evaluation set consisted of 30 one-minute-long video sequences containing a total of 92 individual tracks. Identification performance was evaluated at the track level, and accuracy was used as the primary performance metric.

It is important to note that we were not aiming for frame-wise correct identification, which would have been infeasible, as the lemurs were frequently in positions where identification was impossible (see Fig 5). Instead, we used the tracks to arrive at a majority vote on which lemur is visible in each sequence of detections. This required the model to be confident in situations where the lemur is clearly identifiable and not be overconfident when the decision could not be made. The final decision for which individual is predicted for a track was made by exponentially weighting and summing the confidences for each frame in a majority vote.

We trained the classification branch for individual identification using a ResNet18 [42] model pretrained on the ImageNet [43] dataset. We fine-tuned the network with the individual lemur images, which we cut out from the training images. As ResNet18 expects square images as input, we used squared bounding boxes to avoid squeezing the rectangular bounding boxes that we had previously annotated into squared shapes and introducing biases to the training set.

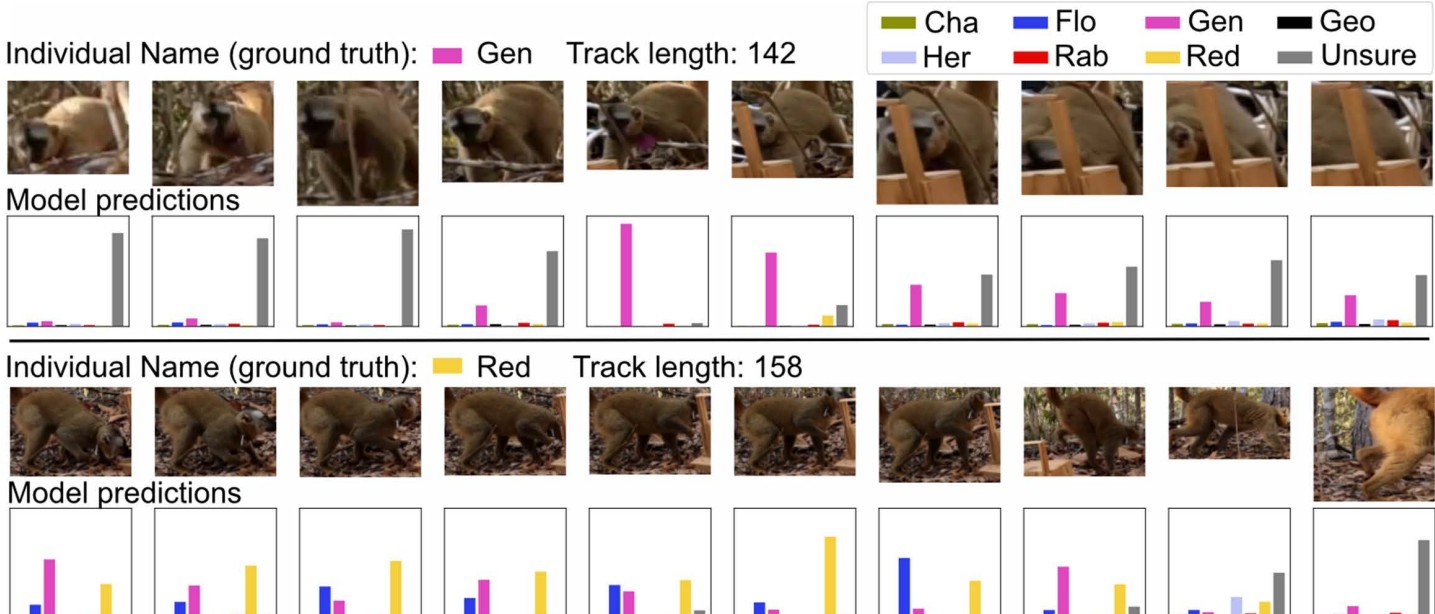

**Fig 5. Two examples that illustrate the difficulty of identifying individuals in every single frame of a track.** Instead, we combine the tracking and identification models to come to a majority vote decision.

We achieved the squared bounding boxes by increasing the size of the shorter sides of the previously annotated bounding boxes to match the longer sides and then cropped and resized them to 224 × 224 pixels. To account for the shift from the perfectly annotated bounding boxes used during training to slightly imperfect bounding boxes predicted during inference, we applied random jitter of up to five pixels to the bounding boxes and random zoom increasing the width and height up to 20% during training. These values were chosen based on a randomized search over the hyperparameter space.

We trained the model with 315 hand-selected images of six individuals. The images were extracted from videos in situations where at least one individual was identifiable, either by their face or collar. After training, we used the model to acquire more training data, by applying it to four additional recordings of experiments. We then created a second training set by randomly sampling 20 frames per track and selecting images manually, only including those that had no errors and were not near duplicates of other selected images. This process resulted in 782 training images. Additionally, we used a small validation set of 35 hand-selected images for selecting the model via early stopping and evaluated the models on 30 one-minute-long snippets.

## Training details

PriMAT was implemented in PyTorch [61] and trained with the Adam optimizer [62]. The training of the base model with MacaquePose (and the MacaqueCopyPaste datasets) was performed on four NVIDIA Quadro RTX 5000 GPUs for two days. We started with a learning rate of $5 \times 10^{-5}$ and reduced it by a factor of 10 after 100 epochs, with a batch size of 8. For fine-tuning with 500 images for both lemurs and macaques, training on one GPU took six hours. The initial learning rate was $10^{-5}$. Since training was performed on images while evaluation was conducted on videos, we saved intermediate model checkpoints to monitor progress and reduce the risk of overfitting.

For the individual identification task, we cropped the images to include only the bounding boxes surrounding lemurs in the dataset and stacked them afterwards, allowing training with a batch size of 128. The model was trained on a single NVIDIA Quadro RTX 5000 GPU, using early stopping with a patience of 10 epochs. Training time with a learning rate of $2 \times 10^{-5}$ was one hour.

Additionally, we trained a model with DeepLabCut [23] version 2.3.8 for which we annotated 33 keypoints per individual. We tested different hyperparameters for pcutoff (confidence for a detection) and IoU. Additionally, we compared DLCRNet_ms5 (the data-driven individual assembly method which was recommended for improved performance in the multi-animal setting) and ResNet50 as backbones. We chose the best performing model with DLCRNet_ms5, pcutoff 0.0 and IoU 0.6. All models were trained for 200,000 training iterations.

We found that the RTX5000 GPUs did not provide sufficient computational capacity to perform inference with the SAM3 model [57]. Consequently, all inference experiments were conducted using A100 GPUs. For each species, we evaluated multiple prompt variations. For lemurs, the prompts included "lemur", "monkey", "primate", "primate without tail", and, for feeding boxes, "wooden box". For macaques, the prompts included "macaque", "monkey", and "primate". We report the range of results obtained across these prompt variations.

## Ethics statement

Both animal studies adhered to the Guidelines for the Treatment of Animals in Behavioral Research and Teaching [63] and the legal requirements of the countries (Madagascar, Thailand) in which the work was carried out.

Elif Karakoç, Richard Vogg, Alexander Ecker, Peter M. Kappeler, and Claudia Fichtel were also involved in the experiment with the redfronted lemurs. The research protocol was approved by the Malagasy Ministry of the Environment, Water, and Forests (permit numbers 064,22/MEDD/SG/DGGE/ DAPRNE/SCBE.Re and 036,24/MEDD/SG/DGGE/ DAPRNE/SCBE.Re), the Mention Zoologie et Biodiversité, Animale de l'Université d'Antananarivo and the CNFEREF Morondava. Data collection in Thailand was authorized by the Department of National Parks, Wildlife and Plant Conservation (DNP) and the National Research Council of Thailand (NRCT) (permit numbers 0401/11121 and 0401/13688).

## Results

With our multi-animal tracking approach, PriMAT, we were able to train models to detect and track individual macaques and lemurs in videos from the wild by only annotating bounding boxes on a few hundred frames. The approach was specifically designed for good performance in primate videos, with solutions for non-standard poses, occlusion, similar appearance, and jumps (see methods). We evaluated the tracking performance of our models with annotated video data from Assamese macaques in Thailand and redfronted lemurs in Madagascar.

### Multi-animal tracking for primates

In this section, we will describe how pretraining influenced model performance for both species, how the choice of different thresholds influences performance and how well the models perform if trained on subsets of the training data.

For pretraining, we experimented with a simple copy-paste strategy for improved generalization (examples see Fig 6A). When evaluating in-domain, we noticed that MacaqueCopyPaste and MacaqueCopyPasteWild showed better results (Fig 6B) than the original MacaquePose [3] dataset, suggesting that these two datasets are "easier" than MacaquePose. Copying and pasting introduces sharper edges and often the animals are more salient against the background, compared to MacaquePose. We also evaluated how well the models performed across domains. While the models trained with Macaque-CopyPaste performed well on all three datasets, MacaquePose showed weaker results in the other domains. MacaqueCopyPasteWild performed well on MacaquePose but poorly on MacaqueCopyPaste (Fig 6C). Both copy-paste approaches outperformed the models trained on MacaquePose when tracking animals on videos from the wild (Fig 6D). Since Macaque-CopyPaste offered greater background variability and matched or exceeded the others in performance, we used it for all subsequent experiments.

We fine-tuned the base models with the 500 annotated images of lemurs and macaques and did an extensive hyper-parameter search for each species (S1 Appendix). Pretraining with ImageNet (70.3 HOTA) was more beneficial than MacaqueCopyPaste (65.7 HOTA) for the lemurs (Table 1). The domain shift from the images of MacaqueCopyPaste to the lemur videos is large, not only because the species look different, but the lemur experiments were filmed from different camera angles. For the macaques however, pretraining with MacaqueCopyPaste (66.6 HOTA) was slightly more beneficial than with ImageNet (64.6 HOTA).

As we did a lot of hyperparameter tuning on the validation videos, we additionally provide 12 test sequences for each species. These can be used for benchmarking other methods against ours in the future. For the lemurs, the backgrounds and lighting conditions vary from experiment to experiment, however, the general setup is similar. The HOTA score on the test set (68.2) is similar to the validation set (70.3). For the macaques, the videos filmed from handheld cameras show higher variability. While the best performing model achieved 66.6 HOTA on the validation set, the score on the test set is 61.6 (see Table 1).

We compared our approach to SAM3 [57], a recent state-of-the-art video tracking foundation model previously shown to perform well on animal tracking tasks [64]. SAM3 performance was sensitive to prompt selection, with alternative prompts leading to reduced tracking accuracy. On the macaque dataset, SAM3 achieved higher tracking accuracy than PriMAT when using the best-performing prompt ("monkey"). In contrast, the prompts "macaque" and "primate" resulted in lower performance than PriMAT. On the lemur dataset, SAM3 failed to reliably segment individuals without their tails, even when using more descriptive prompts. In addition to accuracy, we evaluated computational requirements. SAM3 required substantially greater computational resources and longer runtimes: all experiments were conducted on an NVIDIA A100 GPU, with runtimes exceeding 12 minutes for the macaque dataset and approximately 19 minutes for the lemur dataset, as each class was processed in a separate pass. Peak GPU memory usage reached 14.4 GB and 11.3 GB, respectively. By comparison, PriMAT required less than 5 minutes per species and approximately 5 GB of peak GPU memory (details see Table 1).

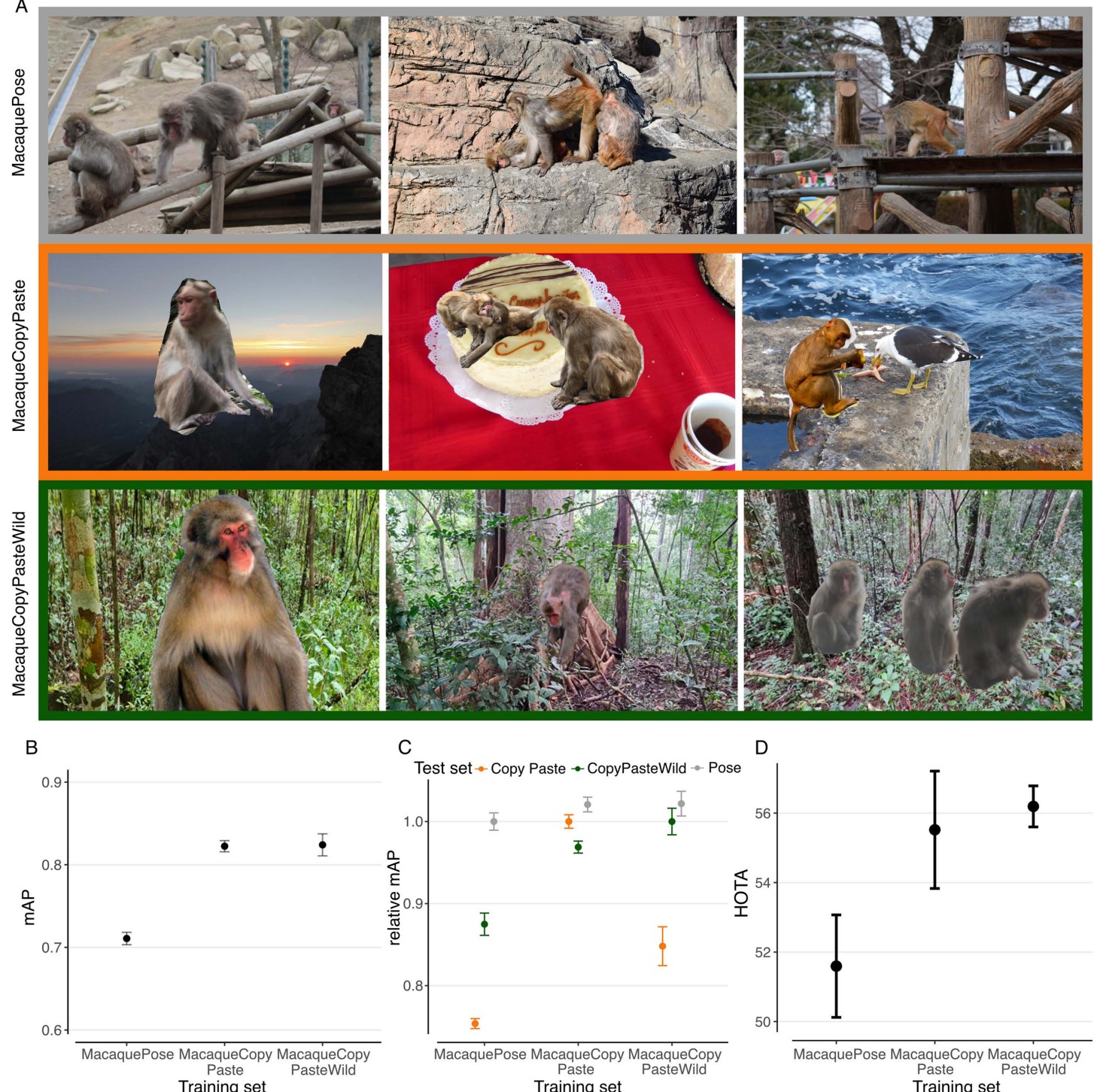

**Fig 6. Pretraining with MacaqueCopyPaste. (A)** Examples from MacaquePose [3] (top), MacaqueCopyPaste with ImageNet [43] backgrounds (middle), and MacaqueCopyPasteWild with backgrounds from Thailand (bottom). **(B)** Models trained on the copy-paste approaches yield higher results when tested *in-domain*, i.e. on their own test sets. This can probably be attributed to sharper edges and more saliency against the background. **(C)** Results of testing on different test sets relative to the in-domain performance from panel B. MacaqueCopyPaste detects macaques well on all three datasets. MacaquePose did not generalize well to other datasets. **(D)** Both copy-paste strategies outperform MacaquePose on the macaque validation videos.

Error bars: Standard error of the mean from training three models with different random seeds. The top row images as well as the monkeys on the other images are reprinted from Labuguen et al. (2020) and Google Open Images under a CC BY 2.0 license. Original copyright remains with the authors. Our model was trained using ImageNet images; however, for illustration purposes in this figure (middle row), all ImageNet backgrounds have been replaced with representative images created by the authors.

**Table 1. Performance of Lemur and Macaque tracking models.**

| | Lemurs | | | | | Macaques | | | | |
|---|---|---|---|---|---|---|---|---|---|---|
| | HOTA | MOTA | IDF1 | Time | VRAM | HOTA | MOTA | IDF1 | Time | VRAM |
| **Validation (Pretraining)** | | | | | | | | | | |
| PriMAT (ImageNet) | **70.3** | **81.5** | **88.1** | | | 64.6 | 75.0 | **83.7** | | |
| PriMAT (MCP) | 65.7 | 67.8 | 79.3 | | | **66.6** | **75.5** | **83.7** | | |
| PriMAT (no pretraining) | 61.3 | 64.8 | 77.0 | | | 59.3 | 62.3 | 75.5 | | |
| **Test** | | | | | | | | | | |
| PriMAT | 68.2 | 74.2 | 83.6 | 04:40 | 0.5G | 61.6 | 69.1 | 79.9 | 04:40 | 0.5G |
| SAM3 ("monkey") | 56.8 | 3.7 | 50.7 | 18:48 | 11.3G | 71.3 | 70.4 | 84.9 | 12:25 | 14.4G |
| SAM3 ("primate") | 52.5 | 22.5 | 49.1 | 18:48 | 11.3G | 58.5 | 43. | 64.9 | 12:25 | 14.4G |
| SAM3 ("lemur"/"macaque") | 51.5 | 26.0 | 48.4 | 18:48 | 11.3G | 55.2 | 6.4 | 65.5 | 12:25 | 14.4G |
| SAM3 ("lemur without tail") | 52.5 | 19.0 | 47.4 | 18:48 | 11.3G | | | | | |

Models were pretrained on different datasets and subsequently finetuned with 500 annotated images of lemurs or macaques. The tracking performance was evaluated on 12 validation video sequences. MacaqueCopyPaste pretraining resulted in improved results in macaque tracking. Pretraining on ImageNet showed better results for lemur tracking. Finally, we report test scores for the best validation setup (pretraining dataset, hyperparameters as detailed in S1 Appendix) for each species on 12 test video sequences. For SAM3 [57], we report several prompts as results differed considerably depending on the input. For lemurs, we additionally prompted "wooden box" for each run.

Additionally, we ran experiments on how the size of the training set influences the performance of PriMAT. For this, we trained models with subsets of the 500 frames to show their usability even with fewer annotated frames (Fig 7). For the lemurs, performance with 100 frames varies between 58.3 and 60.6 HOTA. The performance of the five macaque models trained with 100 frames ranges between 58.1 and 60.2 HOTA. This shows that the training of the bounding box models converges to stable solutions, which do not show a large variance between different subsets.

## Transfer to other species

Applying the model trained jointly on macaques and lemur images without any further training showed good results for the Barbary macaques from Rocamadour, even though the setting in these videos appears quite different from the ones in the videos from Thailand and Madagascar. The model was successful in some cases in detecting baboons, chimpanzees, and gorillas. However, particularly in the PanAf500 dataset, many individuals were not detected. This is likely because the species in this dataset (all great apes) have a very different appearance from lemurs and Assamese macaques, and the camera resolution is much lower than that of our videos. To address this issue, we fine-tuned the model on 100 manually selected frames from the target datasets. We observed high AP50 values, which indicated that most individuals were correctly detected. The lower mAP values indicated that the predicted bounding boxes did not fit perfectly around each individual (Table 2). We show qualitative examples in S1 Appendix.

## Bounding box and keypoint based detection

We did a comparison between our bounding box based lemur model and a keypoint based detection method, namely DeepLabCut (DLC) [23]. First, we observed that only 31% of the visible keypoints were detected by DLC, as many ground-truth

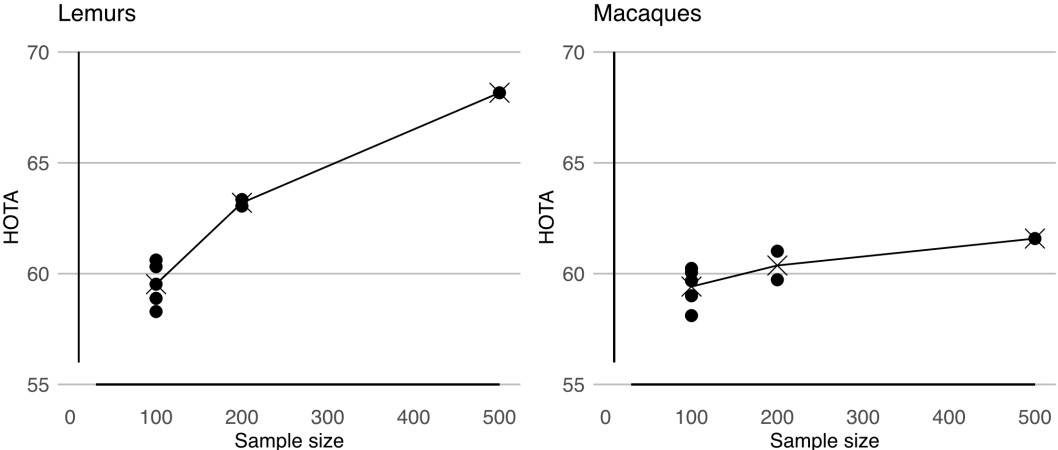

**Fig 7. Showing tracking results for lemurs and macaques on the test videos when the model is trained with a subset of the training data.** We selected five non-overlapping subsets of size 100, and two subsets of size 200. Training with fewer samples will lead to reduced performance, however, we see that even with 100 annotated frames, we get decent tracking results.

**Table 2. Transfer learning results to other species.**

| Dataset | AP50 | mAP |
|---|---|---|
| without training | | |
| Rocamadour | 0.83 | 0.55 |
| Guinea baboons | 0.56 | 0.22 |
| PanAf500 | 0.38 | 0.21 |
| finetuning (100 frames) | | |
| Rocamadour | 0.99 | 0.73 |
| Guinea baboons | 0.77 | 0.41 |
| PanAf500 | 0.81 | 0.47 |

Detection evaluation metrics for models before and after fine-tuning with 100 frames from the target datasets.

skeletons were completely missing or incomplete in the model predictions. Second, among the correctly detected keypoints, the median over the root-mean squared errors of all detections was 49.6 pixels. Of course, this value alone is difficult to interpret, as one pixel represents different distances in different views and we observed large variability between videos (Fig 8B). However, in the lab, DLC models reach median test errors of under 5 pixels with only 100 samples [22] and across different applications [23], which is almost a magnitude less than our model. The relatively large error value from our model was due to the prediction of multiple incomplete skeletons within an individual and skeletons that were split over two or more individuals.

We compared the performance of keypoint models and bounding box models in detecting the presence of a lemur (without punishing for wrong or missing keypoints). Our model outperformed the keypoint based method (Fig 8C/8D). The largest differences were observed in the top view (0.72 vs 0.50) and the close view (0.89 vs 0.72), with minor differences in the far view (0.87 vs 0.83). In one video, which contained high-contrast shadows and therefore a greater need for generalization, the keypoint-based method was not able to detect any keypoints in the whole sequence. The performance of our bounding box method decreased on this challenging video compared to other videos, but it still resulted in a final F1 score of 0.62. For a qualitative comparison, we refer to Fig 9.

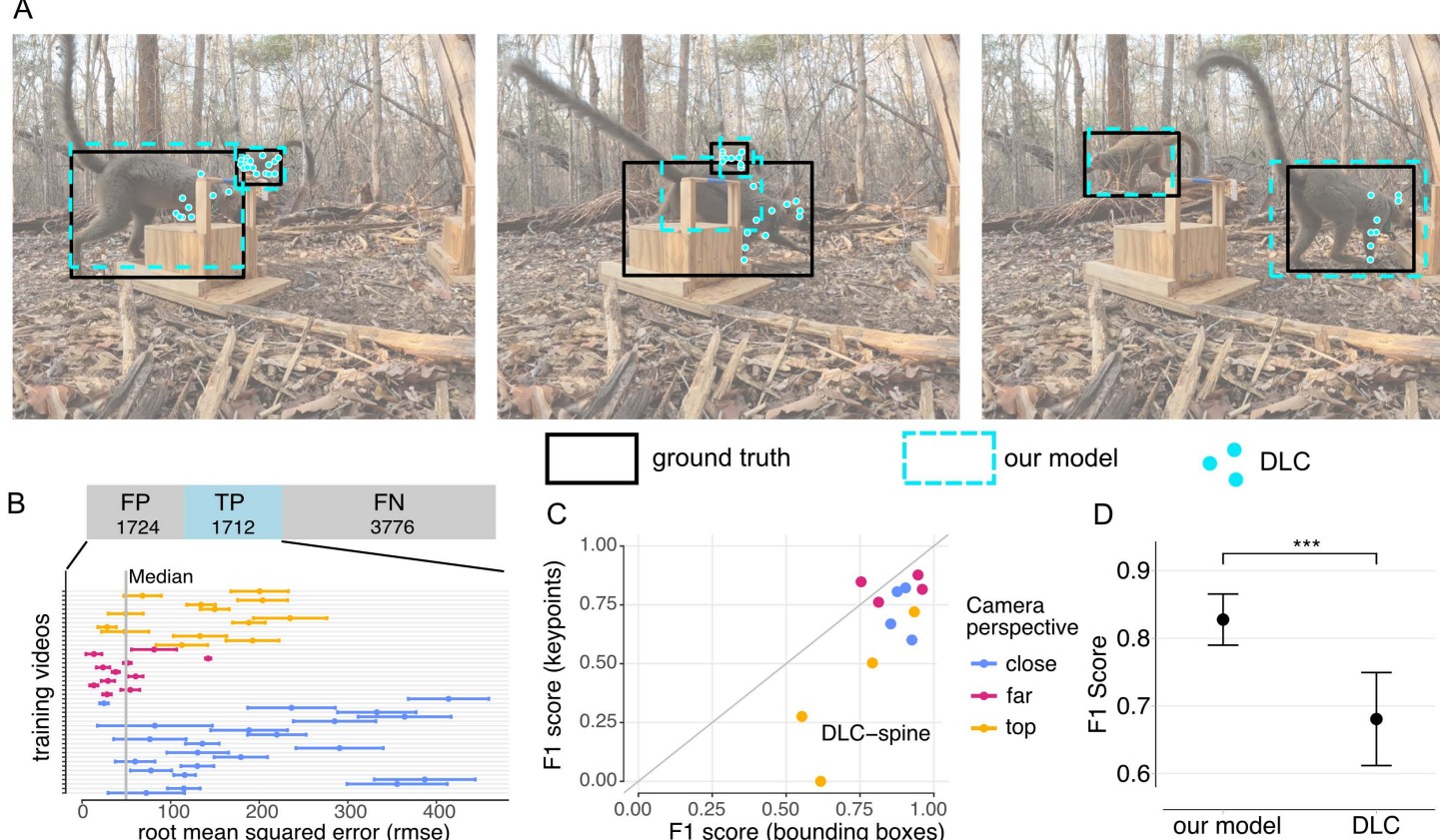

**Fig 8. Comparison between our boxes based tracking model and a DeepLabCut (DLC) model [23].** Our model was trained with 500 images from three camera perspectives. The DLC model was trained using 33 keypoints, resulting in 100 frames. **(A)** A qualitative example where DLC does not detect most keypoints. **(B)** DLC missed many keypoints and shows large root-mean squared errors (rmse) for the predicted keypoints. **(C, D)** Performance difference on the 12 test videos. The evaluation metric was chosen in a favorable manner for DLC, as it counts a detection as positive when there are at least two keypoints correctly detected.

A model that outputs incomplete skeletons, many missing detections or incomplete tracks is of questionable utility for practical applications. To overcome these types of problem and have a robust and generalizing keypoint model, a lot more data is needed.

We performed an additional experiment in which we annotated only seven keypoints per individual and managed to annotate over 200 frames in the allocated five hours. However, this model performed worse than the one with 33 keypoints. This suggests that having a higher number of keypoints with cross-connections is more critical for performance than increasing the number of annotated frames.

### Individual identification

We added a classification branch and applied it to identify individual lemurs in the videos. The final model, which combined tracking and identification information, had an identification accuracy of 84%.

### Discussion

There are many potential paths towards automated behavior analysis using videos of animals in the wild. Most current methods in the lab use keypoint detection and tracking as a starting point, as those provide a valuable intermediate

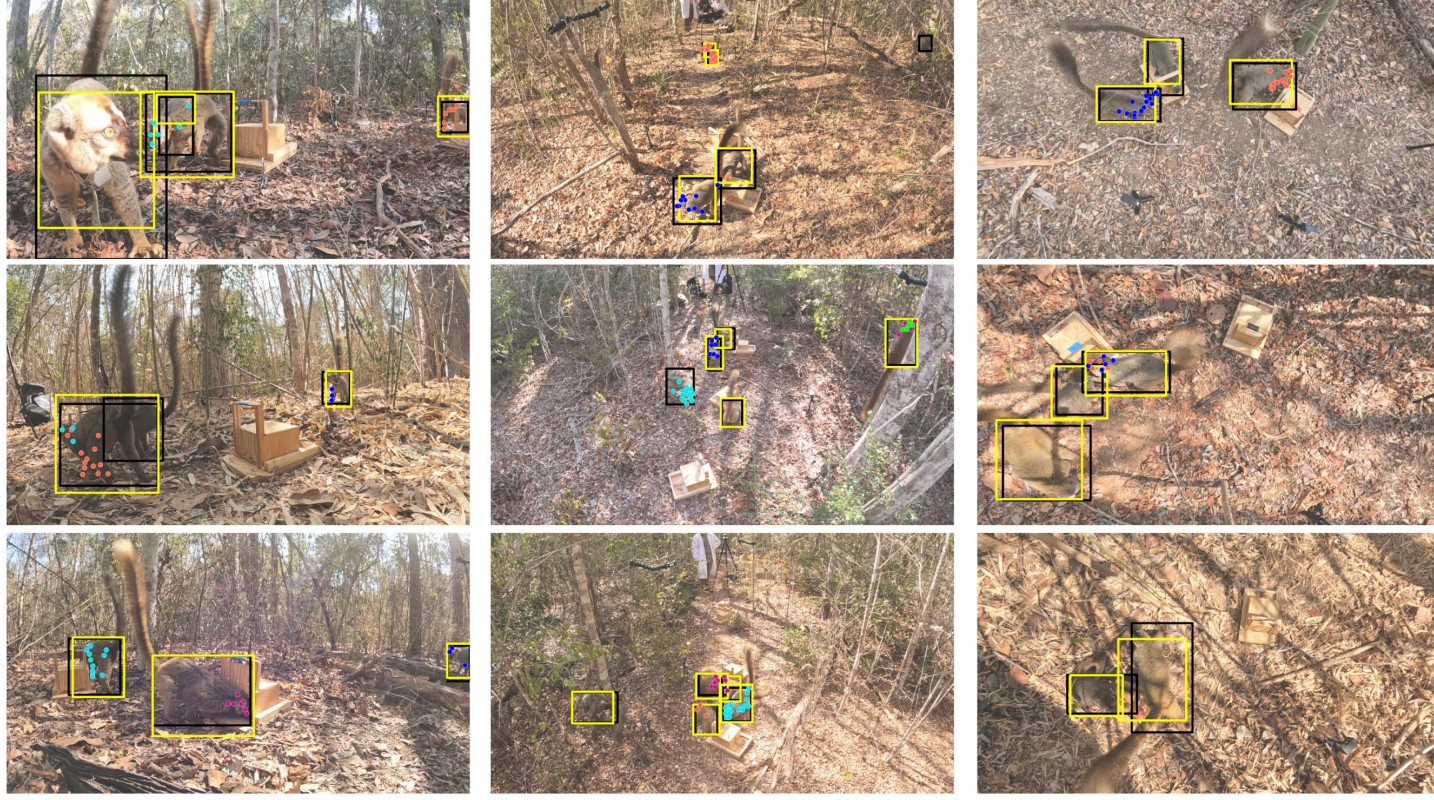

**Fig 9. Qualitative comparison between PriMAT and DLC.** Ground truth bounding boxes are black, PriMAT bounding boxes are yellow. The different colors of the keypoints refer to different detected individuals.

representation, namely the pose of the animal. However, so far, keypoint detection based models have struggled to generalize to highly diverse backgrounds in the wild. One solution could be to annotate vastly larger and diverse data-sets; however, keypoint annotation is a very time-consuming task. Therefore, we proposed an alternative approach based on bounding boxes. We demonstrated that our deep learning based multi-animal tracking approach, PriMAT, outperforms previous methods based on keypoints on videos from the wild while requiring significantly less label-ing time. Our tracking model is based on the FairMOT architecture [36], which was built to track pedestrians. We addressed primate-specific challenges in the design of our approach, including non-standard poses, occlusion, and fast, non-linear motion.

## Key findings and interpretations

We presented several experiments around pretraining on larger datasets and transferring to new settings and species with small annotated datasets. What do our results suggest for researchers working with primate videos from the wild? First of all, pretraining with a general-domain dataset like ImageNet [43] is a good starting point. However, if annotated data from a similar domain as the intended application is available, such as MacaquePose [3] in our case, performance can be improved even further. We showed that copy-paste approaches can be applied to bring the data closer to the domain of application. However, transferring from macaques to lemurs showed a larger domain shift, and in this case, ImageNet pretraining was the better way to achieve good performance. Until a general-domain dataset with many different monkey

species and backgrounds is available, researchers will have to assess each case individually, which pretraining strategies to apply to achieve the best possible performance with a limited labelling budget.

Combining tracking with individual identification models can benefit video analysis, as individuals are often not recognizable on single frames. Using tracks, it suffices to have a few frames with high-confidence prediction and propagate the recognized individual to all the other frames in the track.

## Limitations

Bounding boxes can have drawbacks in certain situations. When elongated objects (such as a monkey with its tail outstretched) are tracked, a large portion of the bounding box can be filled with background. We reduced this problem by excluding the tail from the lemur detections. A second downside is that bounding box coordinates themselves are rarely useful for downstream tasks such as action recognition, which is why subsequently used models always need to process the (cut-out) image information. In contrast, keypoint-based models usually operate directly on the keypoint coordinates and can be more lightweight. Lastly, there are cases when keypoint estimation is needed, for example, when analyzing kinematics [65].

When applying PriMAT to lemur and macaque videos, we observed that the models had difficulties with tracking individuals when there was a lot of fast, non-linear movement and occlusion, such as in videos of juveniles playing. Our association strategy is rule-based and relies on a Kalman filter for linear movement (see S1 Appendix). While adapting the association logic as done in our model can lead to further improvement, more recent multi-object tracking architectures promise to overcome problems with challenging association by learning associations as part of their architecture instead of following hand-crafted rules [66,67]. However, such approaches require densely annotated videos for training, which is more costly than the sparse annotations we use. Hence, their feasibility remains to be evaluated. Another limitation of our approach, shared with many recent tracking methods, is that frames are processed largely independently without explicitly incorporating temporal context. Integrating temporal information could improve performance in many video applications. This limitation may also contribute to differences compared to recent foundation models such as SAM3 [57], which are characterized by large-scale pretraining and text prompts for object detection. While such models can achieve strong performance on well-defined tasks, their performance is sensitive to prompt formulation, as reflected in our results. In addition, SAM3 requires substantially higher computational resources, including GPU memory and inference time. We further observe that these models may be less robust when applied to non-standard or ambiguous objects (e.g., partial animal instances), and cannot easily be finetuned, while PriMAT can be improved by labelling more data. To summarise this comparison, research groups that have access to high-end GPU resources (e.g., A100 or newer) available, that want to track standard objects and not too many object classes and that have a small enough dataset to wait for the inference time, should use SAM3, as it shows remarkably good results in these cases. For all other applications, we see more value in using PriMAT.

In its current form, our classification branch can be used for individual identification only in cases where all identities are known. If there are unknown individuals in the dataset, open-set classification approaches (e.g., via Deep metric learning) [68] are a more promising pathway as they do not require a predefined set of individuals.

## Future directions

While we primarily view the tracking and identification model as a foundation for future action and interaction recognition, it can already offer value in certain use cases, provided a few key prerequisites are met. First, high video quality, including high resolution and a good signal-to-noise ratio, will help to improve performance in both tracking and identification. Second, the degree of background and scene variability influences the amount of training data required; highly different scenes may fall outside the model's domain and reduce performance. Third, the behavior of the primates will determine how well the tracking works. Excessive group movement, such as individuals crossing paths or frequently switching positions, can hinder the ability of tracking

algorithms to reliably distinguish and follow each subject. These prerequisites could, for example, be met in outdoor situations where the camera is filming from within a feeding device, and the question is which individuals are spending how much time at the device, or the camera is filming along a narrow path, and the question is who passes along the path.

PriMAT can be extended to incorporate other tasks, such as instance segmentation [69], and, ultimately, interaction classification [70] between objects when those interactions are detectable in a single frame. The classification branch can be used for individual identification, but also to classify actions performed by the individuals. In the computer vision community, many models developed for spatio-temporal action recognition, interaction recognition or gaze detection build on bounding boxes as intermediate representations [71–74]. The recent datasets PanAf500 [60], BaboonLand [18] or ChimpAct [8] contain activity classes, such as sitting, walking, climbing up or down and would be a suitable benchmark. They could additionally serve as additional pretraining material, such as MacaquePose [3] in our experiments. While our primary focus was on two species of nonhuman primates (Assamese macaques and redfronted lemurs), preliminary qualitative experiments suggest that our approach is similarly applicable to videos of other nonhuman primate species (including Barbary macaques, Guinea baboons, chimpanzees and gorillas). We expect the model to generalize to other animal species with little additional annotation effort.

## Supporting information

**S1 Appendix. Additional figures and tables.** Transfer to other nonhuman primate species and settings, additional technical information on the tracking model and hyperparameter tuning.
(PDF)

## Acknowledgments

We thank the Department of National Parks, Wildlife, and Plant Conservation (DNP) and the National Research Council of Thailand (NRCT) for research permission. We gratefully acknowledge the computing time granted by the Resource Allocation Board and provided on the supercomputer Emmy/Grete at NHR-Nord@Göttingen as part of the NHR infrastructure. The calculations for this research were conducted with computing resources under the project nib00021.

## Author contributions

**Conceptualization:** Richard Vogg, Timo Lüddecke, Zurna Ahmed, Alexander S. Ecker.

**Data curation:** Richard Vogg, Elif Karakoç, Sofia M. Pereira, Suchinda Malaivijitnond, Suthirote Meesawat, Derek Murphy, Julia Fischer, Peter M. Kappeler, Julia Ostner, Oliver Schülke, Claudia Fichtel.

**Formal analysis:** Richard Vogg, Zurna Ahmed.

**Funding acquisition:** Julia Fischer, Florentin Wörgötter, Peter M. Kappeler, Alexander Gail, Julia Ostner, Oliver Schülke, Claudia Fichtel, Alexander S. Ecker.

**Investigation:** Richard Vogg, Matthias Nuske, Timo Lüddecke, Elif Karakoç, Zurna Ahmed, Sofia M. Pereira.

**Methodology:** Richard Vogg, Matthias Nuske, Marissa A. Weis, Elif Karakoç, Zurna Ahmed, Derek Murphy, Alexander S. Ecker.

**Project administration:** Richard Vogg, Alexander S. Ecker.

**Resources:** Julia Fischer, Florentin Wörgötter, Peter M. Kappeler, Alexander Gail, Julia Ostner, Oliver Schülke, Claudia Fichtel, Alexander S. Ecker.

**Software:** Richard Vogg, Matthias Nuske, Marissa A. Weis.

**Supervision:** Julia Fischer, Florentin Wörgötter, Peter M. Kappeler, Alexander Gail, Julia Ostner, Oliver Schülke, Claudia Fichtel, Alexander S. Ecker.

**Visualization:** Richard Vogg.

**Writing – original draft:** Richard Vogg.

**Writing – review & editing:** Marissa A. Weis, Timo Lüddecke, Elif Karakoç, Zurna Ahmed, Sofia M. Pereira, Derek Murphy, Julia Fischer, Florentin Wörgötter, Peter M. Kappeler, Alexander Gail, Julia Ostner, Oliver Schülke, Claudia Fichtel, Alexander S. Ecker.

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
