## [Decision Letter · Decision Letter 0]

17 Nov 2025

PONE-D-25-48058PriMAT: Robust multi-animal tracking of primates in the wildPLOS ONE

Dear Dr. Ecker,

Thank you for submitting your manuscript to PLOS ONE. After careful consideration, we feel that it has merit but does not fully meet PLOS ONE’s publication criteria as it currently stands. Therefore, we invite you to submit a revised version of the manuscript that addresses the points raised during the review process.

Reviewers have now commented on your paper and provide very constructive comments. You will see that they are advising that you revise your manuscript. The reviewers stated that the work is of value, given the new datasets, the evaluation on existing SOTA datasets, and effective visualizations; however, there are several concerns. They show concerns with regard to clearer articulation of the paper’s contributions, particularly about how bounding-box tracking supports behavioral studies in contrast to keypoint-based methods. They emphasize the need for the inclusion of behavior-model evaluation, such as YOLO-Behaviour or X3D, in order to make the argument strong. Also, according to the reviewer, there are problems in the organization, conversational writing style, small dataset size, and a confusing mix of tasks: detection, tracking, behavior, individual ID. They recommend further explanation regarding what the authors mean by individual identification of lemurs, including the role of collars. Moreover, there is mixing and misplacement of ideas in the introduction and methodology sections that need further triangulation. For your guidance, reviewers' comments are appended below.==============================

We look forward to receiving your revised manuscript.

Kind regards,

Dereje Yazezew Mammo, Ph.D.

Academic Editor

PLOS ONE

Journal Requirements:

[This project was funded by the Deutsche Forschungsgemeinschaft (DFG, German Research Foundation) via project number 454648639 – SFB 1528, project number 254142454 – GRK 2070 and project number 502807174 – RTG 2906.

We acknowledge funding by the Leibniz Association through an Audacity Grant from the Leibniz ScienceCampus Primate Cognition (W45/2019 – Strategische Vernetzung).].

[This project was funded by the Deutsche Forschungsgemeinschaft (DFG, German Research Foundation) via project number 454648639 – SFB 1528, project number 254142454 – GRK 2070 and project number 502807174 – RTG 2906. We acknowledge funding by the Leibniz Association through an Audacity Grant from the Leibniz ScienceCampus Primate Cognition (W45/2019 – Strategische Vernetzung). We thank the Department of National Parks, Wildlife, and Plant Conservation (DNP) and the National Research Council of Thailand (NRCT) for research permission. We gratefully acknowledge the computing time granted by the Resource Allocation Board and provided on the supercomputer Emmy/Grete at NHR-Nord@Göottingen as part of the NHR infrastructure. The calculations for this research were conducted with computing resources under the project nib00021.]

[This project was funded by the Deutsche Forschungsgemeinschaft (DFG, German Research Foundation) via project number 454648639 – SFB 1528, project number 254142454 – GRK 2070 and project number 502807174 – RTG 2906.

We acknowledge funding by the Leibniz Association through an Audacity Grant from the Leibniz ScienceCampus Primate Cognition (W45/2019 – Strategische Vernetzung).]

5. We note that you have included the phrase “unpublished data” in your manuscript. Unfortunately, this does not meet our data sharing requirements. PLOS does not permit references to inaccessible data. We require that authors provide all relevant data within the paper, Supporting Information files, or in an acceptable, public repository. Please add a citation to support this phrase or upload the data that corresponds with these findings to a stable repository (such as Figshare or Dryad) and provide and URLs, DOIs, or accession numbers that may be used to access these data. Or, if the data are not a core part of the research being presented in your study, we ask that you remove the phrase that refers to these data.

6. We note that Figure 5 in your submission may contain copyrighted images. All PLOS content is published under the Creative Commons Attribution License (CC BY 4.0), which means that the manuscript, images, and Supporting Information files will be freely available online, and any third party is permitted to access, download, copy, distribute, and use these materials in any way, even commercially, with proper attribution. For more information, see our copyright guidelines: http://journals.plos.org/plosone/s/licenses-and-copyright.

1. You may seek permission from the original copyright holder of Figure(s) [#] to publish the content specifically under the CC BY 4.0 license.

7. Please remove your figures from within your manuscript file, leaving only the individual TIFF/EPS image files, uploaded separately. These will be automatically included in the reviewers’ PDF.

8. We notice that your supplementary figures and tables are included in the manuscript file. Please remove them and upload them with the file type 'Supporting Information'. Please ensure that each Supporting Information file has a legend listed in the manuscript after the references list.

Reviewers' comments:

Reviewer's Responses to Questions

**Comments to the Author**

1. Is the manuscript technically sound, and do the data support the conclusions?

Reviewer #1: Yes

Reviewer #2: Partly

2. Has the statistical analysis been performed appropriately and rigorously? 

Reviewer #1: Yes

Reviewer #2: No

3. Have the authors made all data underlying the findings in their manuscript fully available?

The PLOS Data policy requires authors to make all data underlying the findings described in their manuscript fully available without restriction, with rare exception (please refer to the Data Availability Statement in the manuscript PDF file). The data should be provided as part of the manuscript or its supporting information, or deposited to a public repository. For example, in addition to summary statistics, the data points behind means, medians and variance measures should be available. If there are restrictions on publicly sharing data—e.g. participant privacy or use of data from a third party—those must be specified.requires authors to make all data underlying the findings described in their manuscript fully available without restriction, with rare exception (please refer to the Data Availability Statement in the manuscript PDF file). The data should be provided as part of the manuscript or its supporting information, or deposited to a public repository. For example, in addition to summary statistics, the data points behind means, medians and variance measures should be available. If there are restrictions on publicly sharing data—e.g. participant privacy or use of data from a third party—those must be specified.requires authors to make all data underlying the findings described in their manuscript fully available without restriction, with rare exception (please refer to the Data Availability Statement in the manuscript PDF file). The data should be provided as part of the manuscript or its supporting information, or deposited to a public repository. For example, in addition to summary statistics, the data points behind means, medians and variance measures should be available. If there are restrictions on publicly sharing data—e.g. participant privacy or use of data from a third party—those must be specified.requires authors to make all data underlying the findings described in their manuscript fully available without restriction, with rare exception (please refer to the Data Availability Statement in the manuscript PDF file). The data should be provided as part of the manuscript or its supporting information, or deposited to a public repository. For example, in addition to summary statistics, the data points behind means, medians and variance measures should be available. If there are restrictions on publicly sharing data—e.g. participant privacy or use of data from a third party—those must be specified.

Reviewer #1: Yes

Reviewer #2: Yes

4. Is the manuscript presented in an intelligible fashion and written in standard English?

Reviewer #1: Yes

Reviewer #2: Yes

5. Review Comments to the Author

Reviewer #1: I have recommended minor revisions. You just need to further separate the introduction from the methods section, as some introductory material is in the methods section. You are also missing some important citations, which I have linked.

Reviewer #2: The authors present a fine-tuned model for detecting and tracking non-human primates in camera trap videos. They compare their bounding-box method to more widely used keypoint methods. The authors evaluate their method to publically available primate datasets and release 2 new datasets of lemurs and macaques.

This manuscript could be strengthened by clarifying contribution of this work. The authors assert that bounding box-based tracking is faster to annotation than keypoint detection methods, which is certainly plausible as demonstrated in the experiments and results. However, the authors frame their contribution around behavior studies. Keypoint detections, although more time consuming to annotate, usually generates easy behavior categories "for free" from pose analysis, where bounding boxes do not. I appreciated the authors quantifying the time taken to annotate the videos using their novel method vs key point detection, however their argument for the new method would be strengthened if behavior annotations were added to this analysis as well. I would suggest evaluating the detections on SOTA behavior-identification models, such as YOLO-Behaviour (Chan et al, 2025, MEE) or X3D from BaboonLand. I also suggest revising the manuscript with clearer structure, and clarifying exactly what computer vision tasks are being tackled here - there is a confusing mix of detection, tracking, and individual id, which are all quite different challenges, requiring different evaluation methods.

Strengths:

- novel datasets

- evaluate new method on existing SOTA datasets

- good figures showing the datasets and the challenges with Kalman filters (Fig 3)

Weaknesses:

- Conversational tone of the work is distracting, please revise in more formal writing style

- Manuscript is poorly organized, making the arguments difficult to follow

- No analysis of behavior models is done, although behavior studies are emphasized as an important motivation

- very small dataset

- confusing mix of computer vision tasks - detection, tracking, behavior, and individual id is muddled, making it challenging to evaluate the appropriateness of the metrics reported

- confused about what exactly the authors mean by "individual id" of lemurs - is this truely re-identification of individuals, or just a more robust tracking method? Why is having the collars in view important? Is there a unique marking on the collars? Please clarify this - a figure or photo could help.

6. PLOS authors have the option to publish the peer review history of their article (what does this mean?). If published, this will include your full peer review and any attached files.). If published, this will include your full peer review and any attached files.). If published, this will include your full peer review and any attached files.). If published, this will include your full peer review and any attached files.

...

Reviewer #1: No

Reviewer #2: No

---

## [Author Response · Author response to Decision Letter 1]

21 Feb 2026

We uploaded an external file with responses to the reviewer comments.

Journal Requirements

Regarding 1.

We made sure that our uploaded files meet PLOS ONE's style requirements.

Regarding 2.-4. (Funding information and Financial disclosure)

- We removed the funding information from the acknowledgements. We matched funding information with the financial disclosure statement both in award numbers and order of Funders.

- We would like to add the following Role of Funders statement: „The funders had no role in study design, data collection and analysis, decision to publish, or preparation of the manuscript."

Regarding 5.

- The unpublished data were preprinted in the meantime and we cited the respective study (Pereira et al., 2025).

Regarding 6.

- Former Fig 5 (now Fig 6): the pictures from MacaquePose shown in the top row were published under CC BY 2.0 which allows their use. In the middle and bottom rows we now use only pictures taken by the authors. For the pictures taken by the authors we attach the permissions to publish under CC BY 4.0

Regarding 7.

- We removed the figures from the manuscript and uploaded TIFF files separately.

Regarding 8.

- We removed supplementary figures and tables from the manuscript and uploaded separate files. We ensure that every file has a legend listed in the manuscript.

---

## [Decision Letter · Decision Letter 1]

13 Mar 2026

PONE-D-25-48058R1PriMAT: Robust multi-animal tracking of primates in the wildPLOS One

Dear Dr. Ecker,

Thank you for submitting your manuscript to PLOS ONE. After careful consideration, we feel that it has merit but does not fully meet PLOS ONE’s publication criteria as it currently stands. Therefore, we invite you to submit a revised version of the manuscript that addresses the points raised during the review process.

I would like to thank you for the effort made to address the reviewer comments during your original revision. Both reviewers have provided positive feedback, with one requested a few more small changes before the article is ready for publication. Once answered or corrected the manuscript should be ready for publication. I look forward to seeing the revision submitted and published

Juan

We look forward to receiving your revised manuscript.

Kind regards,

Juan Scheun

Academic Editor

PLOS One

Journal Requirements:

Reviewers' comments:

Reviewer's Responses to Questions

**Comments to the Author**

1. If the authors have adequately addressed your comments raised in a previous round of review and you feel that this manuscript is now acceptable for publication, you may indicate that here to bypass the “Comments to the Author” section, enter your conflict of interest statement in the “Confidential to Editor” section, and submit your "Accept" recommendation.

Reviewer #1: All comments have been addressed

Reviewer #2: All comments have been addressed

2. Is the manuscript technically sound, and do the data support the conclusions?

Reviewer #1: Yes

Reviewer #2: Yes

3. Has the statistical analysis been performed appropriately and rigorously? 

Reviewer #1: Yes

Reviewer #2: Yes

4. Have the authors made all data underlying the findings in their manuscript fully available?

The PLOS Data policy requires authors to make all data underlying the findings described in their manuscript fully available without restriction, with rare exception (please refer to the Data Availability Statement in the manuscript PDF file). The data should be provided as part of the manuscript or its supporting information, or deposited to a public repository. For example, in addition to summary statistics, the data points behind means, medians and variance measures should be available. If there are restrictions on publicly sharing data—e.g. participant privacy or use of data from a third party—those must be specified.requires authors to make all data underlying the findings described in their manuscript fully available without restriction, with rare exception (please refer to the Data Availability Statement in the manuscript PDF file). The data should be provided as part of the manuscript or its supporting information, or deposited to a public repository. For example, in addition to summary statistics, the data points behind means, medians and variance measures should be available. If there are restrictions on publicly sharing data—e.g. participant privacy or use of data from a third party—those must be specified.requires authors to make all data underlying the findings described in their manuscript fully available without restriction, with rare exception (please refer to the Data Availability Statement in the manuscript PDF file). The data should be provided as part of the manuscript or its supporting information, or deposited to a public repository. For example, in addition to summary statistics, the data points behind means, medians and variance measures should be available. If there are restrictions on publicly sharing data—e.g. participant privacy or use of data from a third party—those must be specified.requires authors to make all data underlying the findings described in their manuscript fully available without restriction, with rare exception (please refer to the Data Availability Statement in the manuscript PDF file). The data should be provided as part of the manuscript or its supporting information, or deposited to a public repository. For example, in addition to summary statistics, the data points behind means, medians and variance measures should be available. If there are restrictions on publicly sharing data—e.g. participant privacy or use of data from a third party—those must be specified.

Reviewer #1: Yes

Reviewer #2: Yes

5. Is the manuscript presented in an intelligible fashion and written in standard English?

Reviewer #1: Yes

Reviewer #2: Yes

6. Review Comments to the Author

Reviewer #1: (No Response)

Reviewer #2: The authors present a tracking approach for monkeys and lemurs, representing an important first step toward automated behavior analysis from video. The work demonstrates applicability across multiple species with variable backgrounds and lighting conditions, outperforms keypoint-based tracking while requiring less annotation effort, and includes a preliminary demonstration of individual re-identification on a small lemur dataset.

Primary Concern: The evaluation would be substantially strengthened by a direct comparison against SAM3, the current state of the art for tracking wildlife in videos. This is particularly important given that SAM3 has been evaluated on camera trap video through the SAFARI benchmark,which is closely aligned with the conditions presented in this work. Without this comparison, it is difficult to assess the proposed method's standing relative to the broader field. The authors should either include this comparison or provide a well-reasoned justification for its omission.

Additional Comments:

1. The abstract should explicitly name the four additional species to which the approach was transferred, rather than leaving this implicit.

2. The comparison of annotation effort across techniques (bounding boxes vs. keyframes) is a valuable contribution and is appreciated.

3. It should be clarified why models are trained on images but evaluated on video, this methodological choice warrants explicit justification.

4. The status of the datasets used is unclear. While citations are provided, the dataset repository suggests these may be new to this publication; the authors should explicitly state whether datasets are previously published or newly introduced here.

5. The authors have adequately addressed feedback from the previous review round.

7. PLOS authors have the option to publish the peer review history of their article (what does this mean?). If published, this will include your full peer review and any attached files.). If published, this will include your full peer review and any attached files.). If published, this will include your full peer review and any attached files.). If published, this will include your full peer review and any attached files.

...

Reviewer #1: No

Reviewer #2: **Yes:** Jenna KlineJenna KlineJenna KlineJenna Kline

---

## [Author Response · Author response to Decision Letter 2]

4 Apr 2026

We thank the reviewers for their positive and constructive comments and have updated the manuscript. A detailed response to every comment is uploaded as a pdf file.

---

## [Editor Report · Decision Letter 2]

6 Apr 2026

PriMAT: Robust multi-animal tracking of primates in the wild

PONE-D-25-48058R2

Dear Dr. Ecker,

We’re pleased to inform you that your manuscript has been judged scientifically suitable for publication and will be formally accepted for publication once it meets all outstanding technical requirements.

Kind regards,

Juan Scheun

Academic Editor

PLOS One

Thank you for taking the time to correct all comments and concerns.

I believe your manuscript is now ready
---

## [Editor Report · Acceptance letter]

PONE-D-25-48058R2

PLOS One

Dear Dr. Ecker,

I'm pleased to inform you that your manuscript has been deemed suitable for publication in PLOS One. Congratulations! Your manuscript is now being handed over to our production team.

Kind regards,

on behalf of

Dr. Juan Scheun

Academic Editor

PLOS One